# Monolithically Integrated GaAs Nanoislands on CMOS-Compatible Si Nanotips Using GS-MBE

**DOI:** 10.3390/nano15141083

**Published:** 2025-07-12

**Authors:** Adriana Rodrigues, Anagha Kamath, Hannah-Sophie Illner, Navid Kafi, Oliver Skibitzki, Martin Schmidbauer, Fariba Hatami

**Affiliations:** 1Institut für Physik, Humboldt Universität zu Berlin, Newtonstr. 15, 12489 Berlin, Germany; adriana.rodrigues-weisensee@physik.hu-berlin.de (A.R.);; 2IHP-Leibniz Institut für Innovative Mikroelektronik, 15236 Frankfurt (Oder), Germany; skibitzki@ihp-microelectronics.com; 3Leibniz Institut für Kristallzüchtung, Max-Born-Str. 2, 12489 Berlin, Germany; martin.schmidbauer@ikz-berlin.de

**Keywords:** epitaxial GaAs on Si, nanoheteroepitaxy, defects, Si photonics

## Abstract

The monolithic integration of III-V semiconductors with silicon (Si) is a critical step toward advancing optoelectronic and photonic devices. In this work, we present GaAs nanoheteroepitaxy (NHE) on Si nanotips using gas-source molecular beam epitaxy (GS-MBE). We discuss the selective growth of fully relaxed GaAs nanoislands on complementary metal oxide semiconductor (CMOS)-compatible Si(001) nanotip wafers. Nanotip wafers were fabricated using a state-of-the-art 0.13 μm SiGe Bipolar CMOS pilot line on 200 mm wafers. Our investigation focuses on understanding the influence of the growth conditions on the morphology, crystalline structure, and defect formation of the GaAs islands. The morphological, structural, and optical properties of the GaAs islands were characterized using scanning electron microscopy, high-resolution X-ray diffraction, and photoluminescence spectroscopy. For samples with less deposition, the GaAs islands exhibit a monomodal size distribution, with an average effective diameter ranging between 100 and 280 nm. These islands display four distinct facet orientations corresponding to the {001} planes. As the deposition increases, larger islands with multiple crystallographic facets emerge, accompanied by a transition from a monomodal to a bimodal growth mode. Single twinning is observed in all samples. However, with increasing deposition, not only a bimodal size distribution occurs, but also the volume fraction of the twinned material increases significantly. These findings shed light on the growth dynamics of nanoheteroepitaxial GaAs and contribute to ongoing efforts toward CMOS-compatible Si-based nanophotonic technologies.

## 1. Introduction

Silicon (Si) is the cornerstone of semiconductors in electronics; however, its indirect electronic bandgap limits its application in photonics [1]. In contrast, gallium arsenide (GaAs) has a direct bandgap that enables efficient light emission, making it highly suitable for optoelectronic devices, including lasers and solar cells [2,3,4,5,6]. The monolithic integration of GaAs with large-scale Si represents a major advancement in Si photonics [7,8]. Despite this advantage, achieving high-quality epitaxial growth of GaAs on Si substrate remains a challenge. GaAs and Si have a significant lattice mismatch of 4.2%, more than 50% mismatch in thermal expansion coefficients, and different crystal structures, making the defect-free growth of GaAs on Si difficult [9]. These defects, such as anti-phase domains, misfit dislocations, stacking faults and other structural defects, originate in the early stages of growth [10].

Several approaches have been employed to mitigate defect formation in GaAs layers grown on Si, which arise from strain and lattice mismatch. These include the use of SiGe buffer layers as virtual substrates, selective area growth (SAG), epitaxial lateral overgrowth (ELO), and the aspect ratio trapping (ART) technique [11,12,13,14,15,16].

In SAG and related methods, GaAs is selectively deposited on predefined Si structures using a patterned amorphous mask. A similar approach, nanoheteroepitaxy (NHE), utilizes pre-patterned Si nanotips (NTs) embedded in silicon dioxide (SiO_2_). Unlike other SAG techniques and planar growth methods, NHE distributes strain energy between the epitaxial layer and the Si nanotips, enhancing substrate compliance [17]. NHE has been successfully applied to grow various binary III-V semiconductors, including InP, GaP, and GaN [18,19,20] and it has also been used to grow GaAs on Si NTs using metal organic vapor phase epitaxy (MOVPE) [21,22,23]. The resulting GaAs islands exhibit single-crystalline and nearly relaxed characteristics with a bimodal size distribution. Smaller islands demonstrate high structural quality and well-defined {111}–{100} faceting, while larger islands tend to contain twins [22,23].

In parallel, planar growth studies of GaAs on Si(111) using MOVPE have provided valuable insights into island nucleation and defect evolution. In particular, Miccoli et al. [24] analyzed early-stage GaAs growth on flat Si(111), revealing how island size, faceting, and twin formation evolve during MOVPE. They reported a characteristic bimodal size distribution, where smaller islands exhibit less defects, while larger ones tend to form twins. Although not involving NHE or Si(001), their work offers a relevant reference for understanding morphology–defect correlations in GaAs/Si systems.

In this paper, we present the first demonstration of GaAs nanoheteroepitaxy (NHE) on Si nanotips using gas-source molecular beam epitaxy (GS-MBE), whereas previous reports have relied exclusively on MOCVD-based approaches. Our objectives are to evaluate the suitability of GS-MBE for NHE growth, to identify conditions that enable facet-controlled, low-defect island formation, and to assess their influence on luminescence efficiency. To this end, we conduct a systematic study on samples grown at identical growth rates and temperatures but with varying growth durations. We find that extended growth times, which typically result in larger islands, are associated with a pronounced bimodal size distribution and an increased volume fraction of twinned material. This work provides new insight into the growth dynamics of GaAs NHE via GS-MBE, highlighting the relationship between morphological evolution, facet formation, and defect generation. Our results contribute to advancing the monolithic integration of GaAs on Si and support the development of high-quality III–V-on-silicon materials for optoelectronic and photonic applications.

## 2. Materials and Methods

The selective growth of GaAs islands was performed using GS-MBE on a Si nanotips substrate. The patterned Si NTs were fabricated on 200 mm Si(001) wafers using a state-of-the-art pilot line employing a 0.13 µm SiGe BiCMOS technology. The NTs were arranged in 1.5 cm^2^ square arrays across different regions with varying tip-to-tip distances (pitch size) of 0.5 µm, 0.8 µm, 1 µm and 2 µm. The tops of the Si nanotips were nearly circular, with diameters ranging from 40 nm to 70 nm across the 200 mm wafer. The complete fabrication process of the NTs can be found elsewhere [22]. To minimize the impact of tip size variation on the growth dynamics in our study, we used 2 × 2 cm^2^ wafer pieces with similar arrays as substrates.

Prior to growth, the Si NTs substrates were cleaned using Piranha solution to remove organic contamination. Subsequently, they were dipped in hydrogen fluoride (HF) for 20 s to remove the SiO_2_ layer covering the Si tips, making their tops free of the oxide. The Si tips after etching were uniform, with diameters of approximately 40 nm, as confirmed by atomic force microscopy (AFM). After the cleaning process, the substrates were loaded into the pre-growth chamber of a Riber 32-P GS-MBE system (Riber S. A., Bezons, France) and heated at 200 °C for one hour to eliminate residual water contamination. They were then transferred to the growth chamber, where they underwent an additional annealing step at 720 °C for 5 min to remove any possible native oxide from the Si nanotips. Subsequently, the substrate temperature was reduced to the optimized growth temperature for selective GaAs growth of 580 °C. The deposition of GaAs was initiated using a solid Ga source and thermally cracked arsine (AsH_3_). For all samples, a GaAs deposition rate of approximately 180 nm/h, an AsH_3_ flux of 3 sccm, and a substrate temperature of 580 °C were used. The growth time varied between 1 and 5 h (h). Figure 1 presents a not-to-scale schematic of the key processing steps.

After growth, surface morphology was assessed using scanning electron microscopy (SEM). The evaluation was conducted with the Pioneer Two System by Raith Fabrication operated at 10 kV and 10 mm working distance. The statistical analysis of the size distribution of the islands was performed using the top-view SEM images and a Python 3.12 script.

High-resolution out-of-plane X-ray diffraction (HRXRD) and grazing incidence in-plane X-ray diffraction (GIXD) experiments were carried out on a 9 kW SmartLab system from Rigaku, Tokyo, Japan. For HRXRD, an asymmetric two-bounce Ge 220 collimator was employed along with a parabolic Goebel mirror, providing a primary collimation of about 0.008° at an X-ray wavelength of λ = 1.54056 Å (Cu Kα_1_). Fast reciprocal space maps (RSMs) were recorded by using a HyPix-3000 area detector with 100 × 100 μm^2^ pixel size. The GIXD investigations were performed at glancing angles of incidence close to the critical angle of total external reflection of Si (α_c_ = 0.22°) to ensure maximum sensitivity to the GaAs islands and to suppress the signal from the underlying Si substrate. In order to investigate crystal twinning, complementary X-ray pole figures were recorded at the GaAs 111 Bragg reflection at the full polar (0°–90°) and azimuthal (0°–360°) angular range with a receiving detector slit of 0.5°.

Photoluminescence (PL) measurements were performed using a 520 nm laser diode (L520P50) from ThorLabs (Newton, NJ, USA) with a laser power density of 540 mW/cm^2^. The Maya2000Pro-NIR (Oceanoptics, Orlando, FL, USA) spectrometer covering a range between 780 and 1200 nm. The sample was placed in a SHI-4-2 closed-cycle helium cryostat (JANIS Research CO, USA) with a vacuum-jacketed cold head and recirculating helium gas, enabling temperature-dependent measurements down to 5 K. The system was operated using a Sumitomo Heavy Industries HC-4E2 helium compressor (Sumitomo Heavy Industries, Tokyo, Japan).

## 3. Results and Discussion

First, we focus on the growth condition for the selective epitaxy of GaAs on Si nanotips. We initially determined the optimal growth window, defined by the substrate temperature, AsH_3_ flux, and Ga growth rate. This optimized range ensures the appropriate conditions for the selective nucleation of GaAs on the Si tips. The nanoheteroeptiaxy of III-V compounds requires lower growth rates compared to thin-film growth, promoting adatom diffusion to the tips and their nucleation. The adatoms either adsorb on the surface and begin to diffuse or undergo the re-desorption process. The growth temperature needs to be sufficiently high to prevent the sticking of GaAs on the SiO_2_ mask. Otherwise, parasitic growth of GaAs across the entire surface occurs. In order to determine the optimized growth temperature, we grew a series of samples with a Ga rate of about 180 nm/h and AsH_3_ flux of 2.3 sccm, while varying the growth temperature. The samples were subsequently analyzed using SEM to evaluate the selectivity of the growth. Under the aforementioned conditions, the determined growth temperature ranges from 520 °C to 585 °C.

We now turn our discussion to the influence of the growth time on size and morphology using top-view SEM images. To eliminate the effect of pitch size, we compare samples with the same tip-to-tip distance of 1 µm. For the smaller pitch sizes and extended growth times, the islands are interconnected. Figure 2 presents top-view SEM images of samples grown for 2 h, 3 h, 4 h, and 5 h. For clarity in the discussion, we refer to the samples by their respective growth times.

In order to understand the dynamics involved in the growth and to obtain a quantitative size distribution, a statistical analysis was performed. The size distribution was analyzed using top-view SEM images of the islands, processed by a Python script designed to determine the area of the islands using the findContours function of the OpenCV library. Additional details can be found in reference [19]. Top-view SEM images of each sample, containing hundreds of GaAs islands were supplied as inputs to the program and the top-view area of the islands was analyzed and put into histograms. Corresponding area histograms, and the numerical fits for each sample are presented in Figure 2.

The SEM images clearly assert the successful development of a selective GaAs growth process, displaying that each individual Si nanotip is fully covered by a single GaAs island. The histograms obtained from 2 h and 3 h samples can be fitted very well using a single Gaussian function, with a peak position at 0.013 µm^2^ ± 0.006 µm^2^, and 0.039 µm^2^ ± 0.013 µm^2^, respectively. The islands of these samples are characterized by a uniform size distribution with an equivalent diameter of 130 nm ± 30 nm, and 220 nm ± 35 nm, respectively. The equivalent diameter is calculated as the radius of a disc with the same projected lateral area of an island.

With increasing growth time, and hence nominal thickness, two types of islands form. Figure 2e exhibits a SEM top-view image of the 4 h sample. Some islands exhibit a rectangular shape and smaller size, referred to as mode 1 (highlighted by a red square), while others are large size and have multiple facets, referred to as mode 2 (highlighted by a blue square). The histogram and the related fit for the islands of 4 h sample are shown in Figure 2f. The best fit is using two log-normal distributions, with peak positions at 0.11 µm^2^ ± 0.038 µm^2^ and 0.16 µm^2^ ± 0.038 µm^2^, as depicted in Figure 2f. According to the fit, the islands for mode 1 possess an average equivalent diameter of 380 nm ± 65 nm, while for mode 2 the diameter is 440 nm ± 50 nm.

A top-view SEM image of the 5 h sample is shown in Figure 2g. Red and blue squares highlight modes 1 and 2, respectively. A bimodal size distribution is again observed, but it is more pronounced compared to the 4 h sample. The size distribution of the islands’ top-view area, along with the corresponding fit, is also presented in Figure 2g. The curve is fitted using two log-normal distributions, with peak positions at 0.21 ± 0.11 µm^2^ and 0.46 ± 0.11 µm^2^. The average equivalent diameters are 520 ± 130 nm for mode 1 and 760 ± 90 nm for mode 2.

To evaluate the degree of bimodality, we calculated the overlap ratio, which is defined as the ratio of the integral over the region where the distributions of mode 1 (represented by the red curve) and mode 2 (represented by the blue curve) overlap to the integral of the entire distribution. The complement of this overlap ratio, referred to as the bimodal separation coefficient, quantifies how distinctly the two modes are separated within the bimodal distribution. A higher bimodal separation coefficient indicates a greater distinction between the two modes.

For the 4 h and 5 h samples, the bimodal separation coefficients are 0.88 and 0.95, respectively. This clearly demonstrates that increasing the nominal thickness of GaAs results in the bimodal size distribution with a more pronounced distinction between the larger and smaller islands.

Figure 3a presents a plot of the average island top area for mode 1 and mode 2 as a function of growth time. The bars, indicating the tolerance range, estimated from the full width at half-maximum (FWHM) of the fit curves in the histograms. For growth times shorter than 4 h, a monomodal growth behavior is observed, whereas longer growth times lead to a bimodal growth pattern.

We now turn our attention to the morphology of the islands. Figure 3b presents a 45°-tilted SEM image of the 5 h sample. The islands clearly display varying shapes, multiple facets, and different heights, which complicates the accurate determination of their average height and the aspect ratio.

A high-magnification top-view SEM image of two GaAs islands from the 2 h sample is shown in Figure 4a. In these islands, four different facet orientations are identified, corresponding to {001} planes. Figure 4b presents a high-magnification top-view SEM image for the 5 h sample. The red square highlights the smaller island belonging to mode 1, while the blue square marks the larger island, belonging to mode 2. The larger island exhibits multiple crystallographic facets, making a precise indexing just using the image impossible.

In order to obtain further information about the crystal structure and strain state of the GaAs islands, high-resolution X-ray diffraction measurements were performed. Figure 5 shows out-of-plane reciprocal space maps in the vicinity of the out-of-plane asymmetrical 224 Bragg reflection for all samples. In addition to the sharp and intense peak originating from the Si substrate (denoted as S), a broad intensity feature originating from the GaAs islands (denoted as F) can be identified. The intensity of F is substantially reduced for samples with smaller growth times, which can be attributed to the reduced amount of material deposited, resulting in lower diffraction intensity.

The mean position of the island peaks F with respect to the substrate reflection S throughout indicates a cubic symmetry of the GaAs lattice, which implies that the GaAs islands are fully relaxed for all samples. It is noteworthy, however, that the F peaks are remarkably elongated in angular direction, i.e., perpendicular to the scattering vector, while they are quite sharp in radial direction, i.e., along the scattering vector. This observation indicates an angular distribution of net-planes within the GaAs islands, while the local cubic lattice symmetry is preserved. A quantitative analysis leads to a lattice parameter of a_GaAs_ = 5.656 (5) Å, which is in excellent agreement with reported values of bulk GaAs (a = 5.653 Å), confirming the fully relaxed zinc-blende crystalline structure.

Despite the signal of the GaAs islands in the vicinity of the 224 reciprocal lattice vector being extended in the angular direction, no evidence of crystal twinning in the GaAs islands is provided by the reciprocal space maps. However, a preliminary indication of twinning can be observed in a grazing-incidence in-plane X-ray diffraction 2θ-ϕ-scan (see Figure 6) conducted along the Si [110] substrate direction. Similar to previous results of MOVPE grown GaAs NHE islands, a 111 GaAs Bragg peak is observed [23]. This is indicative of secondary twinning, as the twinned <111> directions are in close proximity (approximately 3.7°) to the un-twinned [110] direction. Due to the limited instrumental resolution, this difference is small enough that the corresponding intensity from the twinned net-planes can also be detected.

In order to further investigate the twinning process occurring within the GaAs islands, X-ray pole figures were measured. Figure 7a–d illustrates the intensity distributions recorded at 2θ = 27.1°, corresponding to the GaAs 111 Bragg reflection, for samples grown for 2 h, 3 h, 4 h and 5 h, respectively. A corresponding simulation of the pole figure considering single twinning is shown in Figure 7e. In this simulation, we consider rotational twins formed by a single 60° rotation around the <111> directions [25]. In addition, the simulation also accommodates the 4-fold symmetry of the Si substrate, resulting in four equivalent in-plane orientations of the GaAs nanoislands, each rotated by 90° around the surface normal. As a result, the simulated pole figure exhibits a fourfold symmetry.

A comparison between experiment and simulation reveals that the four prominent peaks (surrounded by blue circles) observed at a polar angle of approximately 54.7° are indicative of un-twinned regions within the GaAs islands. Beyond this, a substantial variety of additional spots are observed. The yellow marked peaks in the simulation are caused by single twinning. As these peaks are observed for all pole figures in Figure 7a–d single twinning is present for all samples, although the pole figures of the 2 h and 3 h samples (Figure 7a and Figure 7b, respectively) show comparatively low counting statistics, which is due to the low amount of material. Conversely, even very weak intensity spots can be detected for the 4 h and 5 h samples, which do not appear in the simulation in Figure 7e and which correlate with the presence of multiple higher-order twinning. These are formed by combined rotations around the [111], [111], and [111] directions as already described in [25].

In order to quantify the single-twinning effect as a function of growth time, the integrated intensity of selected spots in the pole figure was evaluated. The four inner peaks in the pole figure at polar angle of 15.8° (marked by yellow circles) are caused by single-twinned areas in the GaAs islands, while the four spots at polar angles of 54.7° (marked by blue circles) are caused by un-twinned areas in the GaAs islands. The integrated intensity ratio of both parts as a function of growth time is illustrated in Figure 7f. Since the pole figures were measured under identical experimental conditions, the intensity ratio is proportional to the volume fraction of single-twinned and un-twinned material inside the GaAs islands.

It is evident that the intensity ratio tends to increase with higher growth time, indicating that the sample grown for 5 h has the highest volume fraction of twinned material. It is important to note that the intensity ratio does not directly measure the number of twins but rather provides an estimate of the overall volume fraction of twinned material as compared to un-twinned material. Notably, for the 5 h sample, which exhibits bimodal growth, the volume fraction is twice that of the 2 h sample, which shows monomodal growth.

Twinning occurs when the crystal lattice undergoes mirror symmetry across a plane, often as a mechanism for strain relaxation during island growth or coalescence. While HRXRD results indicate that GaAs islands are, on average, fully relaxed over an ensemble, strain distribution may remain non-uniform. Localized strain fields, particularly near interfaces, can persist, influencing defect formation and island morphology.

The optical properties of the samples were investigated using photoluminescence spectroscopy. Figure 8a,b presents the PL spectra of the ensemble of islands for the 5 h sample (0.8 µm pitch size) and the 4 h sample (1 µm pitch size), measured at 7 K. Corresponding top-view SEM images are displayed on the right. For reference, the PL spectrum of a Si nanotip wafer is also included (Figure 8c). Multiple photoluminescence (PL) peaks are observed, all of which lie below the fundamental bandgap at the Γ-point of relaxed bulk GaAs (1.52 eV at 7 K). The PL spectra of our GaAs islands grown on Si tips exhibit dominant features that closely resemble those reported for epitaxial GaAs on planar Si(001) substrates at low temperature, including a prominent excitonic emission near 1.489 eV and a characteristic valence-band splitting due to residual strain [26]. These emission energies are significantly redshifted compared to fully relaxed GaAs. We attribute this shift—similar to the case of epitaxial GaAs on Si—to residual tensile strain that develops during cooldown, caused by the mismatch in thermal expansion coefficients between GaAs and Si. Although the GaAs islands are largely relaxed at room temperature due to their size and three-dimensional geometry, differential contraction upon cooling introduces a tensile strain component that reduces the bandgap energy through potential deformation effects.

We therefore assign the highest intensity peak at 1.479 eV to strain-shifted heavy-hole recombination in GaAs, involving free and/or bound excitons [27]. The nearby peak at 1.472 eV is attributed to carbon acceptor-related transitions under similar tensile strain. Defect-related features are also evident, with emissions at 1.417 eV and 1.432 eV attributed to structural defects at the GaAs/Si interface in combination with carbon acceptors [26,28]. In addition to the main excitonic features, a weaker peak is observed around 1.504 eV, which we associate with regions of the larger GaAs islands that are less affected by thermal strain—most likely the upper parts of the islands where relaxation is more complete. The coexistence of these emissions indicates the presence of a strain gradient within the islands, underscoring the complex interplay between geometry, strain relaxation, and thermal expansion mismatch in shaping the optical response. A broad, persistent feature centered at 1.24 eV is attributed to multiple defect-related transitions, potentially involving diffusion-induced defects [28].

In the case of the 5 h sample with a 0.8 µm pitch size (Figure 8a), we observed larger islands—some even beginning to coalesce—compared to the 1 µm pitch samples (see Figure 2g). The increased island size and onset of coalescence may explain the appearance of additional PL signals between 1.24 and 1.4 eV, which suggest the formation of defects and optically active states, resulting in the plateau-like feature in the PL spectrum. These findings point to the importance of understanding the impact of growth dynamics and defect formation on the optical properties of the grown islands.

Our results demonstrate that the size distribution, faceting, and defect formation—particularly twinning—are interconnected phenomena that influence the luminescence properties of the GaAs islands. These characteristics are driven by growth kinetics, strain, and surface energy minimization. The observed bimodal size distribution of GaAs islands during nanoheteroepitaxy on Si nanotips can be attributed to a combination of diffusion-driven coarsening and geometry-induced growth effects. As the islands grow and their separation becomes comparable to or smaller than the diffusion length of Ga adatoms (typically 100 nm to 1 µm, depending on temperature and V/III ratio), material exchange between islands becomes possible. In this regime, Ostwald ripening can occur, leading to material transfer and further size disparities [29]. This phenomenon is observed in the 4 h and 5 h samples and aligns with previous observations on NHE-grown GaAs islands by MOVPE [23], where bimodal size distribution, faceting and twin formation were found to be more pronounced in larger islands. Larger GaAs structures tend to develop well-defined crystallographic facets to minimize surface energy over extended surface areas. Similar faceting behavior has been observed in other GaAs nanostructures, such as nanowires and nanoislands, grown by various selective area epitaxy techniques [23,30,31,32,33]. Larger structures often have more defects, as their size and surface area enable multiple relaxation pathways. Among such defects, twins can significantly influence faceting by locally altering strain fields and surface energies. Interestingly, it has also been reported that using Si doping, twin formation and faceting in selectively grown GaAs nanowires can be increased significantly [33]. These twin defects have been shown to affect the growth kinetics of specific low-index GaAs facets, ultimately contributing to the emergence of tetrahedral facet terminations. This further supports the interplay between defect formation, twinning, and morphological evolution in the nanoheteroepitaxy of GaAs islands.

## 4. Conclusions

In summary, we report the first demonstration of GaAs nanoheteroepitaxy on a Si(001) CMOS-compatible nanotip wafer using GS-MBE, expanding the range of viable techniques for NHE beyond previously reported MOCVD-based approaches. Through detailed morphological, structural, and optical analyses—employing scanning electron microscopy, X-ray diffraction, and photoluminescence—we observed a transition from monomodal to bimodal growth as the islands increased in size with longer growth times. Larger islands exhibited multiple crystallographic facets, while an extended growth duration led to a higher volume fraction of twinned material within the GaAs islands. This suggests a direct correlation between bimodal growth, facet formation, and twin evolution. Our findings underscore the intricate relationship between size, strain, and surface energy in determining the structural and morphological evolution of GaAs islands.

However, further studies are needed to elucidate the underlying growth mechanisms in more detail. In particular, systematic control over facet formation, twin suppression, and dopant incorporation will be critical for achieving high-quality III-V nanoislands suitable for device integration. Advanced in situ and ex situ characterization, such as synchrotron-based X-ray nanodiffraction or cathodoluminescence mapping, could provide deeper insights into local strain relaxation and defect distributions. Looking ahead, our results open new opportunities for the monolithic integration of GaAs-based optoelectronic devices on Si, such as nanolasers, LEDs, or photodetectors. Optimizing island uniformity, defect density, and emission wavelength tunability will be essential steps toward practical applications in silicon photonics and on-chip light sources.

## Figures and Tables

**Figure 1 nanomaterials-15-01083-f001:**
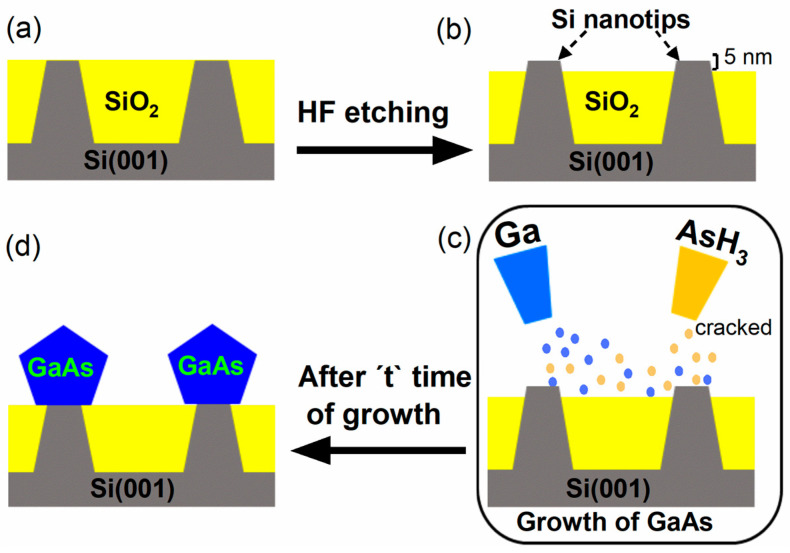
Schematic of the steps involved in the selective growth of GaAs islands on Si(001) nanotip substrate: (**a**) The cross-sectional view of the initial substrate, with Si nanotip (gray) arranged in a SiO_2_ matrix (yellow). (**b**) Top-free Si nanotips after HF etching. (**c**) The wafer in the growth chamber supplied with Ga and thermally cracked AsH_3_ for the growth process. (**d**) Cross-sectional view of the GaAs islands (blue) grown selectively on Si NTs.

**Figure 2 nanomaterials-15-01083-f002:**
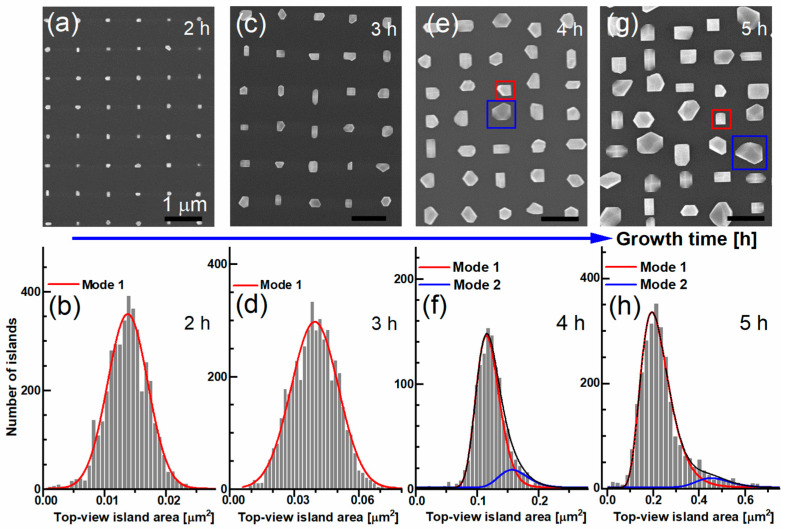
(**a**) Top-view SEM image of GaAs islands selectively grown on Si nanotips after 2 h of growth, showing various island shapes. (**b**) Area histogram of the islands and its fitted curve after 2 h growth. (**c**) Top-view image of the sample after 3 h of growth. (**d**) Histogram of the islands and the fitting for sample 3 h. (**e**) A top-view SEM image of GaAs islands after 4 h of growth and (**f**) corresponding area histogram and fits. Red square highlights the smaller islands (referred to as mode 1), while the blue square indicates the larger islands (referred to as mode 2). (**f**) The area histogram of the islands after 4 h of growth, with two log-normal fits representing the size distributions for the small and large islands. (**g**) Top-view image of the islands after 5 h of growth, (**h**) the histogram and two log-normal fitted curves. The blue arrow marks increasing growth time, indicating the island sizes evolution over time. All samples were grown at similar conditions at 580 °C and with a nominal growth rate of 180 nm/h.

**Figure 3 nanomaterials-15-01083-f003:**
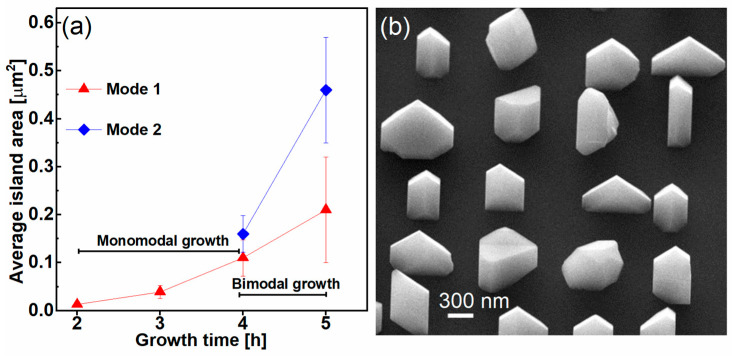
(**a**) Plot of average island top area for mode 1 and mode 2 as a function of growth time, obtained from the histograms in Figure 2b,d,f,h, where the ranges of monomodal and bimodal growth are highlighted; (**b**) 45°-tilted SEM image of the 5 h sample.

**Figure 4 nanomaterials-15-01083-f004:**
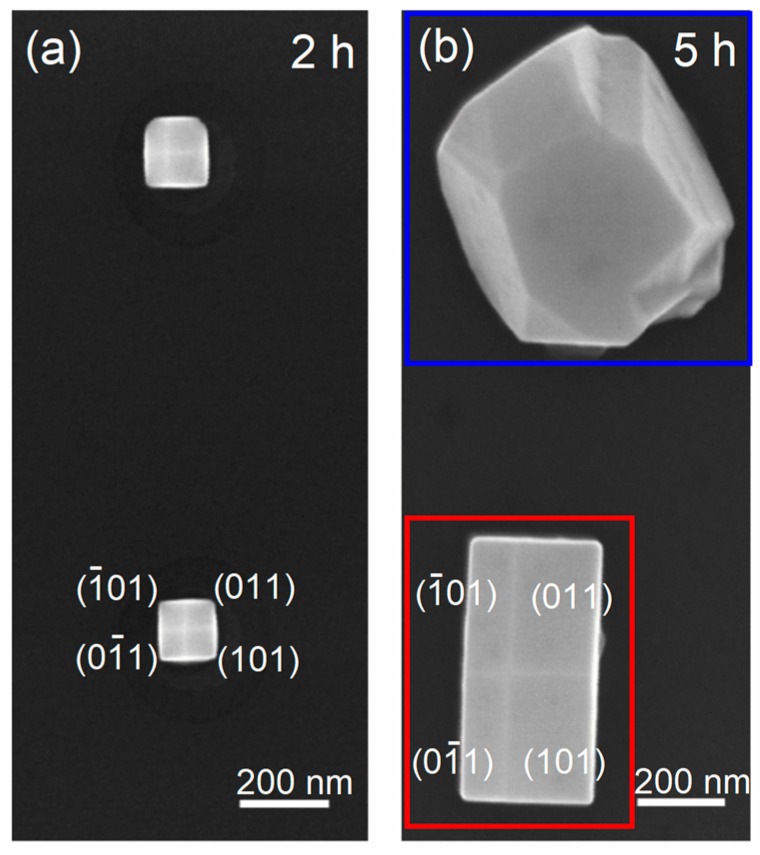
High-magnification top-view SEM images of GaAs islands grown on Si NTs showing the top facets: (**a**) 2 h sample and (**b**) 5 h sample. The red square highlights the smaller island belonging to mode 1 while the blue square indicates the larger island, belonging to mode 2.

**Figure 5 nanomaterials-15-01083-f005:**
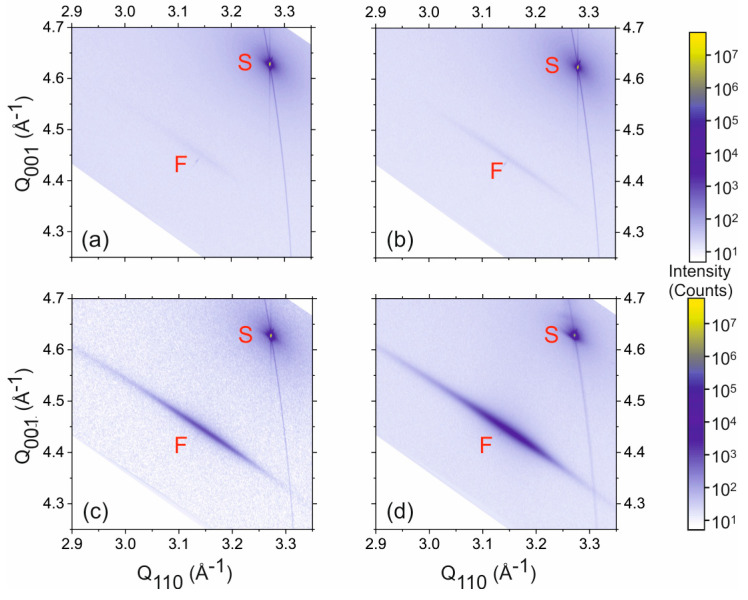
X-ray reciprocal space maps in the vicinity of the 224 asymmetric Bragg reflections obtained for the samples with GaAs islands (**a**) 2 h sample, (**b**) 3 h sample, (**c**) 4 h sample, and (**d**) 5 h sample. The Si substrate and GaAs island peaks are marked as ‘S’ and ‘F’, respectively. The color bars represent the diffraction intensity in counts.

**Figure 6 nanomaterials-15-01083-f006:**
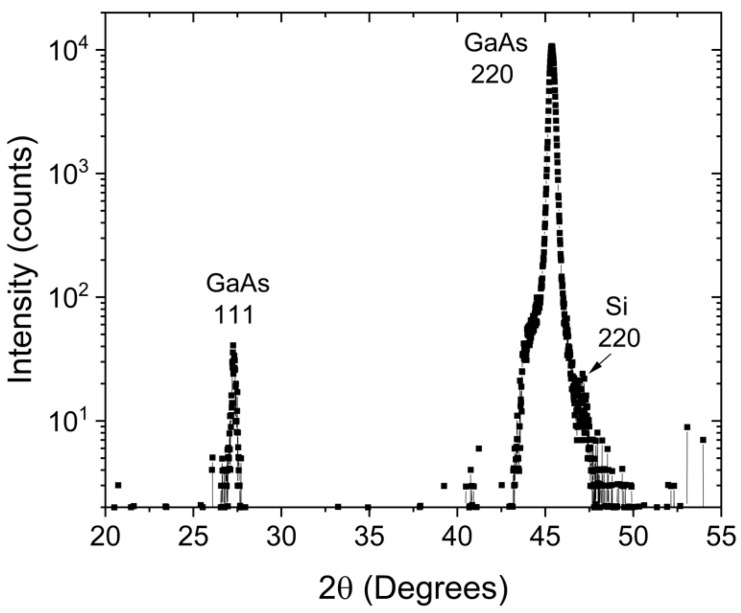
Grazing-incidence in-plane X-ray diffraction 2θ-ϕ-scan along the [110] direction for the 5 h sample. Besides the strong GaAs 220 Bragg peak caused by un-twinned GaAs a weak 111 GaAs, a Bragg peak is observed, which is caused by secondary twinning. The weak peak on the right side of the GaAs 220 reflection is due to the Si 220 Bragg reflection.

**Figure 7 nanomaterials-15-01083-f007:**
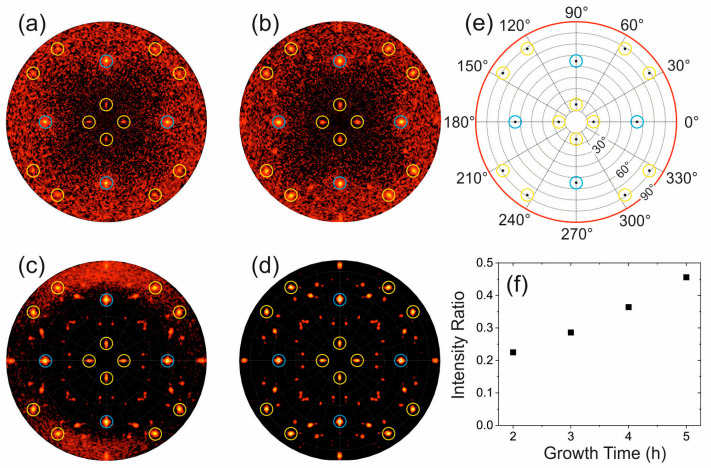
X-ray pole figures recorded at 2θ = 27.1° (GaAs 111 Bragg reflection) for the samples grown with different growth times: (**a**) 2 h, (**b**) 3 h, (**c**) 4 h and (**d**) 5 h, respectively, along with (**e**) simulations considering single twinning. Blue circles indicate un-twinned material, while yellow circles mark single twinned material. (**f**) Ratio of integrated intensity of peaks at a polar angle of 15° (inner yellow circles; single twins) and of peaks at polar angle of 54.7° (blue circles; un-twinned) versus growth time.

**Figure 8 nanomaterials-15-01083-f008:**
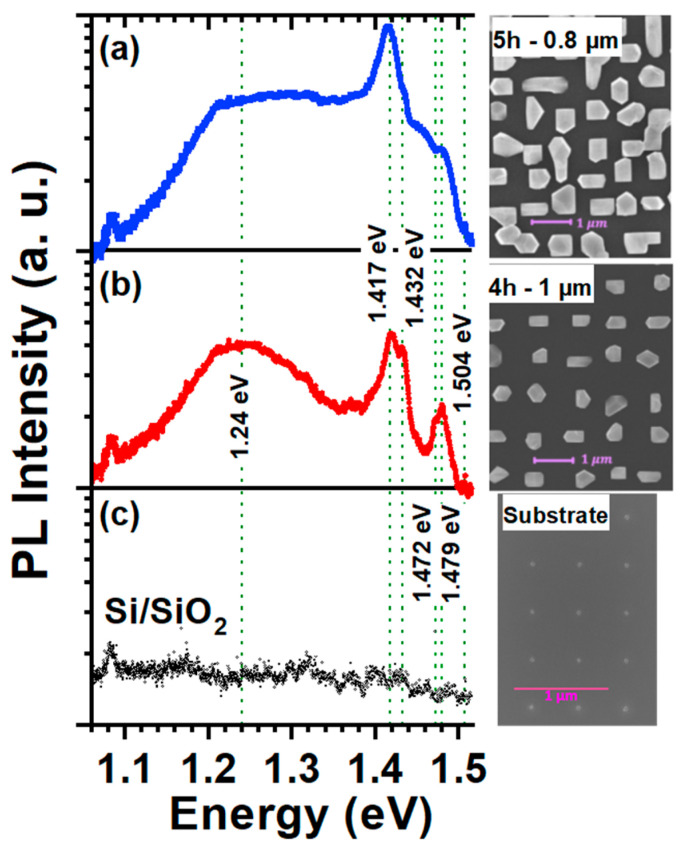
PL spectra of the 5 h (**a**) and 4 h (**b**) samples with pitch sizes of 1 µm and 0.8 µm, respectively, measured at 7 K and laser power density of 540 mW/cm^2^. The corresponding top-view SEM images are shown. For reference, the measured spectrum of the Si nanotip substrate (**c**) without GaAs islands is included.

## Data Availability

Data are contained within the article.

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
