# Peer review of "Monolithically Integrated GaAs Nanoislands on CMOS-Compatible Si Nanotips Using GS-MBE"

_nanomaterials, 2025, doi:10.3390/nano15141083_

Round 1
Reviewer 1 Report
Comments and Suggestions for Authors
The Article is firn. However, some modifications are required.
- The abstract should be rewritten by removing the abbreviation from it.
- The figure caption should be rewritten in detail.
- Does the growth temperature play any role in the Selective area growth?
- State-specific study objectives at the end of the Introduction.
- Compare PL results with additional literature and discuss defect peaks.
- Add “CMOS” and “BiCMOS” to the abbreviations list.
- It is suggested that provide the line scan of the EDAX measurement because there is no direct measurement to show that on the outside of the crystal, there is no material growth.
- Fix grammatical errors.
- Add recent references about the epitaxial growth by MBE to strengthen the Introduction. Articles (Advanced Materials Interfaces, 8 (14) (2021), 2100593; ACS Applied Optical Materials 2024, 2, 7, 1353–1359) can be useful to fellows.
Author Response
Comments 1: The abstract should be rewritten by removing the abbreviation from it. |
Response 1: We hank the reviewer for pointing this out. We have removed it from abstract (lines 10, 12 and 15).
|
Comments 2: The figure caption should be rewritten in detail. |
Response 2: We hank the reviewer for this suggestion. We added more details in the figure captions.
Comments 3: Does the growth temperature play any role in the Selective area growth? Response 3: We appreciate the reviewer’s question. In selective area growth, particularly, in nanoheteroepitaxy the deposition area for III-V materials is strongly sensitive to the growth temperature. To clarify this point, we have added new information on page 4, lines 137-150.
Comments 4: State-specific study objectives at the end of the Introduction. Response 4: We thank the reviewer for pointing this out. We have modified the introduction accordingly on page 2, line 59-79.
Comments 5: Compare PL results with additional literature and discuss defect peaks Response 5: We appreciate the reviewer’s suggestion. The PL spectra are discussed more extensively in the revised version, supported with references on page 10-11, lines 337-360).
Comments 6: Add “CMOS” and “BiCMOS” to the abbreviations list. Response 6: We have added these two terms to the abbreviations.
Comments 7: It is suggested that provide the line scan of the EDAX measurement because there is no direct measurement to show that on the outside of the crystal, there is no material growth. Response 7: We thank the reviewer for the suggestion. In response, we performed EDX measurements on representative samples. The images in the attached file show surface maps for Ga and As after 5 hours of GaAs growth. These clearly confirm that no growth occurs on the SiOâ‚‚ mask between the islands. However, we believe that the high-resolution SEM images already included in the manuscript — both planar and tilted views — provide strong and sufficient evidence for the selectivity of the growth. In all samples, no GaAs is observed on the SiOâ‚‚ mask, as shown in Fig. 2, 3, and 4.
Comments 8: Fix grammatical errors. Response 8: We acknowledge this comment and did a careful proofread to fix the errors.
Comments 9: Add recent references about the epitaxial growth by MBE to strengthen the Introduction. Articles (Advanced Materials Interfaces, 8 (14) (2021), 2100593; ACS Applied Optical Materials 2024, 2, 7, 1353–1359) can be useful to fellows. Response 9: We thank the reviewer for this suggestion and carefully reviewed the publications titled “Influence of Temperature on Photodetection Properties of Honeycomb-like GaN Nanostructures” and “A Self-Driven Bidirectional Photocurrent Photodetector for Optically Controlled Logic Gates Utilizes a GaN-Nanowall Network”. While both contributions are highly interesting, we found it challenging to establish a direct link between their focus and our study on GaAs nanoheteroepitaxy using gas-source MBE. If the suggestion was based on the use of plasma-assisted MBE, we respectfully note that this growth technique differs in several key aspects from our NHE GS-MBE approach. Given the different material systems and growth mechanisms, we feel that including these references may not directly support the discussion in our manuscript. We hope the reviewer agrees that maintaining a focused reference list helps to keep the manuscript concise and relevant.
|

Reviewer 2 Report
Comments and Suggestions for Authors
This article reports on the investigation of the selective growth of fully relaxed GaAs nanoislands on CMOS-compatible Si(001) nanotip wafers using gas-source molecular beam epitaxy via a nano-heteroepitaxy approach. The manuscript focuses on understanding the influence of the growth conditions on the morphology, crystalline structure, and defect formation of the GaAs islands.
The topic is interesting. However, the manuscript must be revised, and the following main points must be carefully addressed before it can be considered for publication:
- It is important that the authors clarify, inside the manuscript, what is the added value of this article to the field of research among the existing literature on the topic.
- The Introduction is poor and needs to be enlarged/completed by extending further the discussion on the state-of-the-art in the field. For example, for completeness, it is worth mentioning comprehensive research on the shape, size evolution and nucleation mechanisms of GaAs nanoislands such as the one reported in this relevant paper that deserve to be cited/discussed [https://doi.org/10.1021/acs.cgd.9b00225].
- The text reported between line 116 and line 151 is duplicated and exactly repeated at lines 153-188. The text at lines 116-151 must be completely deleted because it is not appropriate for that paragraph.
- Regarding the description of the instrumentation used, it is necessary to report the brand and model and parameters used in the measurement of each instrument used for PL measurements, such as cryostat, monochromator, laser, detector, etc. (diffraction grating used in the monochromator, slit width and relative resolution, integration time of the measurements, laser power density, and so on).
- Regarding the comment of Fig.2, the average diameters need to be reported with the relative errors and the standard deviations.
- In addition to the nanoisland size distribution, the nanoisland aspect-ratio should be calculated from Fig 3 (b) and critically discussed in the text.
- Regarding the X-ray pole figures reported in Fig. 7, the authors mention the appearance of multiple higher order twinning without explaining in detail their rotation origin. To complete the information and discussion on this fundamental point, the authors should consider and comment previous results recently reported in this pertinent paper that should be mentioned [https://doi.org/10.1016/j.apsusc.2023.157627].
- Regarding the interpretation of PL measurements, the authors mention the thermal strain as the cause of energy shift of the peaks, but with the size of the nanoislands (average diameters between 380 nm and 760 nm and high more than 300 nm, from Fig. 3 b) the nanoislands are fully relaxed, as also stated in the abstract (line 12) and in the text (line 255 and following) as demonstrated by the X-ray reciprocal space maps. Consequently, the interpretation of the abovementioned PL results is certainly wrong and needs to be correct. In addition, the authors ascribe the peaks at 1.479 eV and 1.472 to exciton transitions and valence band splitting (lines 334-335) but, considering the band gap of 1.519 eV at 7K, the energy difference of these features and the gap, being of 40-47 meV, is huge for attribution to exciton transitions and valence band splitting. Also this point should be properly corrected and carefully clarified in the text.
- The conclusions should be completed with a better and more in-depth outline of possible future developments/perspectives.
- All bibliographic references must be completed with the DOI. In addition, the list of bibliographic references shows the reference number repeated, once as a simple number, and once again the same number in square brackets.
Author Response
Comments 1: It is important that the authors clarify, inside the manuscript, what is the added value of this article to the field of research among the existing literature on the topic. |
Response 1: We thank the reviewer for this important comment. We have revised the Introduction and Conclusion to better highlight the added value of our study on page 2, 11, and 12, lines 59-79, 386-388, and 399-407. In particular, we now emphasize that this work demonstrates, to the best of our knowledge, the first application of gas-source molecular beam epitaxy (GS-MBE) for GaAs nanoheteroepitaxy (NHE) on Si. While one or two reports have explored NHE using MOCVD, the GS-MBE approach has not yet been addressed in the literature. The NHE concept offers a promising pathway for strain management and defect reduction by relaxing lattice mismatch at the nanoscale. In this context, our systematic study provides insight into the morphological evolution of GaAs islands under varying growth durations, including the onset of bimodal size distributions and facet development. We believe this work advances the understanding of NHE-based integration strategies and demonstrates the viability of GS-MBE for monolithic III–V/Si integration.
|
Comments 2: The Introduction is poor and needs to be enlarged/completed by extending further the discussion on the state-of-the-art in the field. For example, for completeness, it is worth mentioning comprehensive research on the shape, size evolution and nucleation mechanisms of GaAs nanoislands such as the one reported in this relevant paper that deserve to be cited/ discussed [https://doi.org/10.1021/acs.cgd.9b00225]. |
Response 2: We thank the reviewer for flagging this issue and comment. We have modified the introduction on page 2, lines 59-65 and added the new reference [Ref 24].
Comments 3: The text reported between line 116 and line 151 is duplicated and exactly repeated at lines 153-188. The text at lines 116-151 must be completely deleted because it is not appropriate for that paragraph. Response 3: We appreciate the reviewer for pointing out this oversight. We deleted the duplicated text.
Comments 4: Regarding the description of the instrumentation used, it is necessary to report the brand and model and parameters used in the measurement of each instrument used for PL measurements, such as cryostat, monochromator, laser, detector, etc. (diffraction grating used in the monochromator, slit width and relative resolution, integration time of the measurements, laser power density, and so on). Response 4: We thank the reviewer for pointing this out. We have revised this part on page 4, lines 128 -135.
Comments 5: Regarding the comment of Fig.2, the average diameters need to be reported with the relative errors and the standard deviations. Response 5: We have addressed the reviewer's comment by revising the text and including the standard deviation for each mean diameter and the peak position on page 4,5 lines 167-190.
Comments 6: In addition to the nanoisland size distribution, the nanoisland aspect-ratio should be calculated from Fig 3 (b) and critically discussed in the text. Response 6: We are grateful to the reviewer for raising this point. In contrast to the work of Miccoli et al. [Ref24], the islands in our samples exhibit varying shapes and facet orientations, making average height and aspect ratio less meaningful as comparative parameters across different growth conditions. To clarify this, we have revised the paragraph on page 6, lines 221-222.
Comments 7: Regarding the X-ray pole figures reported in Fig. 7, the authors mention the appearance of multiple higher order twinning without explaining in detail their rotation origin. To complete the information and discussion on this fundamental point, the authors should consider and comment previous results recently reported in this pertinent paper that should be mentioned [https://doi.org/10.1016/j.apsusc.2023.157627] Response 7: We would like to thank the reviewer for this valuable comment and the reference to the paper that we were not aware of. We generally agree with the reviewer that we should describe the twinning process in more detail. We have therefore first explained the basis of our simulations – for single twinning - in more detail on page 8, lines 279-287 and have also included the mentioned paper as a new reference [Ref. 25]. Moreover, although multiple high-order twinning is not in the main focus of the present paper, we have also included on page 9 lines 296-298 a sentence about the rotation origin of multiple higher order twinning. A more detailed discussion of multiple high-order twinning for our samples would certainly be interesting, but is beyond the scope of this paper.
Comments 8: Regarding the interpretation of PL measurements, the authors mention the thermal strain as the cause of energy shift of the peaks, but with the size of the nanoislands (average diameters between 380 nm and 760 nm and high more than 300 nm, from Fig. 3 b) the nanoislands are fully relaxed, as also stated in the abstract (line 12) and in the text (line 255 and following) as demonstrated by the X-ray reciprocal space maps. Consequently, the interpretation of the abovementioned PL results is certainly wrong and needs to be correct. In addition, the authors ascribe the peaks at 1.479 eV and 1.472 to exciton transitions and valence band splitting (lines 334-335) but, considering the band gap of 1.519 eV at 7K, the energy difference of these features and the gap, being of 40-47 meV, is huge for attribution to exciton transitions and valence band splitting. Also this point should be properly corrected and carefully clarified in the text. Response 8: We thank the reviewer for this important observation and the opportunity to clarify the origin of the PL features. We agree that, as stated in the manuscript and demonstrated by X-ray reciprocal space mapping, the GaAs nanoislands are largely relaxed at room temperature. However, the PL measurements were performed at low temperature (7 K), where residual tensile strain can be reintroduced due to the difference in thermal expansion coefficients between GaAs and Si. Specifically, upon cooldown, Si contracts less than GaAs, leading to tensile stress within the GaAs islands, particularly in regions near the interface with the substrate. This thermally induced strain has been reported in GaAs/Si epitaxial planar heterostructures (e.g., Ref [26]) and leads to a reduction in bandgap energy that can explain the observed redshift in the PL peaks. Regarding the attribution of the PL peaks near 1.479 eV and 1.472 eV: we acknowledge the reviewer’s concern that the ~40–47 meV energy difference from the relaxed GaAs bandgap at 7 K (1.519 eV) may be too large for pure excitonic transitions or valence-band splitting alone. However, we interpret these emissions as strain-shifted transitions, involving either free or bound excitons (in the 1.479 eV peak), and transitions involving impurity-related levels, such as the carbon acceptor (1.472 eV), both under the influence of tensile strain. The energy shifts introduced by this thermally induced tensile strain can reach tens of meV depending on the local strain magnitude and position within the island, consistent with previous reports on strained GaAs on Si. Furthermore, the coexistence of a higher-energy peak at ~1.50 eV suggests inhomogeneous strain relaxation within the islands. To clarify this in the manuscript, we have revised the relevant section (lines 337-360) to emphasize the role of cooling-induced tensile strain, distinguish it from the room-temperature relaxation state, and to reframe the interpretation of the 1.479 eV and 1.472 eV peaks as strain-shifted transitions involving both excitonic and impurity-related recombination.
Comments 9: The conclusions should be completed with a better and more in-depth outline of possible future developments/perspectives. Response 9: We acknowledge and thank the reviewer for flagging this point, and we have made appropriate changes in the revised manuscript on page 11-12, lines 399-407.
Comments 10: All bibliographic references must be completed with the DOI. In addition, the list of bibliographic references shows the reference number repeated, once as a simple number, and once again the same number in square brackets. Response 10: We appreciate the reviewer’s suggestion. We have corrected the reference numbers and added the DOI for all references (see the References list).
|
Round 2
Reviewer 1 Report
Comments and Suggestions for Authors
Accept
Author Response
Please see our Round 1 response!
Reviewer 2 Report
Comments and Suggestions for Authors
The authors have properly addressed all issues, and the manuscript is now eligible for publication in the journal.
Author Response
Please see our Round 1 response!